# Maternal Parenting and Preschoolers’ Psychosocial Adjustment: A Longitudinal Study

**DOI:** 10.3390/ijerph192113750

**Published:** 2022-10-22

**Authors:** Nicla Cucinella, Rossella Canale, Maria Valentina Cavarretta, Sonia Ingoglia, Nicolò Maria Iannello, Cristiano Inguglia

**Affiliations:** 1Department of Psychology, Educational Science and Human Movement, University of Palermo, 90128 Palermo, Italy; 2Department of Law, University of Palermo, 90134 Palermo, Italy

**Keywords:** parenting stress, positive parenting, preschool, social–emotional competence, emotional–behavioral difficulties

## Abstract

Previous research reported that positive parenting and parenting stress might impact children’s psychosocial adjustment. The current longitudinal study aimed at evaluating the associations over time between mothers’ positive parenting, their parenting stress, and their preschoolers’ social–emotional competence and emotional–behavioral difficulties. Participants were 53 Italian mothers, aged between 24 and 47 years (M = 35.30, SD = 5.28) at T0, and their children (females = 51%), aged between 3 and 6 years (M = 4.48, SD = 0.84) at T0. Mothers completed self-report scales at 2 time points (with a 2-year lag). An autoregressive cross-lagged model was tested that had a good fit to the data, χ^2^(6) = 3.37 ns, CFI = 1.00, RMSEA = 0.00. The results showed that maternal positive parenting at T0 negatively predicted maternal parenting stress at T1; maternal parenting stress at T0 negatively predicted children’s social–emotional competence at T1. Moreover, at each time point, children’s social–emotional competence was associated positively with maternal positive parenting and negatively with maternal parenting stress; children’s emotional–behavioral difficulties were positively associated with maternal parenting stress. The results confirm that interactions with mothers are fundamental for children’s psychosocial adjustment. Implications for research and practice aimed at reducing parenting stress and fostering positive parenting are discussed.

## 1. Introduction

Positive child psychosocial adjustment is reflected in the development of adequate social–emotional competence, which is considered by various researchers to be a fundamental index of personal and interpersonal well-being [1,2]. For example, children with higher social–emotional competence possess greater prosociality and empathy [3], are considered positively by teachers and peers [4], and show better social problem-solving skills [5]. On the contrary, deficits in social–emotional competence have been associated with childhood psychopathology [6], risk of academic failure [7], delinquent behavior and/or substance addiction in late adolescence [8]. Social–emotional competence involves skills that fall under both the emotional domain, such as the ability to understand their own and others’ emotions and to regulate and to express them appropriately [9], and the social domain, such as the ability to achieve personal goals in interactions with others while simultaneously maintaining positive social relationships in different situations and over time [10,11]. Consequently, the elements of social–emotional competence are important contributors to a child’s successful and effective interaction, allowing her/him to engage in sustained and positive interactions with peers, marked by favorable and regulated emotions [2]. According to this view, social–emotional competence assumes crucial importance especially in preschool age (from 3 to 6 years), as this period of life represents a sensitive period of development, which lays the foundations for the acquisition of those behaviors, attitudes and capacities which are the basis of interactions and relationships among peers [12]. Children with lower social–emotional competence, in fact, are more at risk for the manifestation of peer relational problems and emotional–behavioral difficulties [13].

Another indicator of child psychosocial adjustment that is closely related to social–emotional competence is in fact emotional–behavioral difficulties. These are typically grouped in the literature into two broad categories: externalizing and internalizing problems; externalizing problems include aggressive behavior, hyperactivity and impulsivity, while internalizing problems include symptoms that are indicative of emotional suffering, like anxiety, depression and social withdrawal [14,15]. Children with early externalizing problems show significantly poorer school readiness and social skills [16] and are disadvantaged in the transition to elementary school as they have difficulties adapting to the new environment and to the school routine [17]. Furthermore, children with early internalizing problems tend to be less socially competent and exhibit higher rates of social problems [18]. Therefore, it is clear that the emotional–behavioral difficulties experienced by preschool children can undermine their positive development and have a serious impact on their social and psychological well-being, especially in the long term. Previous longitudinal studies have indeed found that early-onset and co-occurring emotional–behavioral difficulties tend to follow a stable or increasing pattern over time, predicting increased odds of psychiatric diagnoses in late adolescence as well as decreased occupational opportunities and income [19,20].

Thus, identifying the psychological factors that could promote or compromise preschoolers’ psychosocial adjustment in terms of social–emotional competence and emotional–behavioral difficulties is the key to curbing the long-term negative sequelae that derive from potential social, emotional and behavioral dysregulation in early years and that can lead to psychological ill-being during childhood and adolescence. From this perspective, several studies [21,22,23,24] have identified some family factors that can be associated with children’s psychosocial adjustment in terms of social–emotional competence and of the emotional–behavioral difficulties.

### 1.1. Maternal Positive Parenting and Children’s Psychosocial Adjustment

Among these family factors, mothers’ parenting behaviors seem to play a central role in promoting the positive development of children, in particular their social–emotional competence [25]. Indeed, it is within the family environment—thanks to daily dyadic interactions with mothers or other caregivers—that children have the first chance to learn the ability to build social relationships with others [26], by learning to regulate and to express their emotions appropriately [27].

Positive parenting in particular seems to significantly influence children’s psychosocial adjustment, contributing to a broad range of social, emotional and behavioral outcomes [28]. According to Flannery et al. [29], rather than being conceptualized as the absence of harsh or negative parenting behaviors, positive parenting represents an independent parenting dimension that should be considered separately from harsh parenting. It is a relatively wide dimension that reflects mother’s warmth and involvement, vocal positivity and praise, responsiveness to her child’s affective and physical cues and needs, and sensitive tuning with her child’s capabilities and to the developmental tasks he/she faces [29,30,31]. 

Mothers who use positive parenting behaviors tend to react to their children’s needs during daily dyadic interactions, talking to the child, engaging the child in shared entertainment or encouraging the child to explore the environment [32]. These behaviors are associated with multiple indicators of children’s self-regulation. More sensitive mothers with their infants from 3 to 7 months of age have toddlers who are more regulated at 12–14 months [33]; mothers who express more warmth with their toddlers at 20 months have more regulated children at 34 months [34]; and mothers with more sensitivity and warmth towards their 30-month-old child predicted his/her regulation, in the form of effortful control, at 42 months [35]. In a sample of German kindergartners, maternal warmth was related to children’s behavior regulation during a snack delay task, and mothers’ contingent response to distress was related to children’s respect for the rules [23]. In a study examining 252 mother–child dyads, early maternal warm responsiveness at 12 months of child’s age had a direct effect on child social skills at 54 months [36]. Furthermore, numerous studies have found associations between positive parenting, especially in the form of maternal warmth, and fewer externalizing and internalizing problems in children [24,37,38]. 

### 1.2. Maternal Parenting Stress and Children’s Psychosocial Adjustment

In sum, the quality of parenting has been shown to be the main vehicle for fostering children’s psychosocial well-being [39]. However, since parenting is a condition subject to challenges deriving from the care, protection and education of children [40], it represents an extremely complex and potentially very stressful task that often has to be carried out within demanding situations, with limited personal resources and in relation to a child who might be very difficult to manage [41,42]. 

The construct of parenting stress refers to a specific form of stress experienced by parents that arises from the demands of being a parent, more specifically from a perceived discrepancy between situational demands related to parenting and personal resources [43,44]. When parenting stress reaches a clinically significant level, it may lead to negative parenting, such as lower responsiveness in childcare [45]; limited involvement and less warmth in interactions with children [46]; attitudes of rejection [47] or coercive, controlling and punitive educational practices [48]. 

In general, high levels of maternal parenting stress have been consistently associated with problematic child development. Children of mothers with high levels of parenting stress often report major social, emotional and behavioral problems, including anxiety-depression, social withdrawal, attention difficulties, aggression and rule-breaking, during preschool and school age [49]. In more recent studies, Planalp et al. [50] found a correlation between greater parenting stress in a group of adolescent mothers and child’s lower self-regulation at 18 months; Santelices et al. [51] and Cucinella et al. [52] showed that parenting stress in mothers was related to greater difficulties in social–emotional development in their preschool children. Moreover, in a sample of Swedish mothers of 436 children with an average age of 7, a higher level of parenting stress was associated with greater externalizing and internalizing problems and with lower social competence in children [22]. Finally, in a longitudinal study [21] that examined the relationships between postpartum parenting stress in 682 first-time mothers and their children’s social–emotional competence at age 3, results indicated that relatively higher initial levels of maternal parenting stress and its increase over the years predicted lower social–emotional competence in children; specifically, mothers who reported high stress levels early after childbirth and increasing stress over time later had children with lower levels of prosocial functioning (i.e., more behavior problems). 

From these results, it would therefore appear that maternal parenting stress plays a relevant role in negatively influencing child psychosocial adjustment. Nevertheless, the findings linking maternal parenting and child adjustment have to be interpreted with caution because although research has consistently supported the idea that parents influence their children’s behavior and adjustment, some authors [53,54] have examined the opposite hypothesis, which is that children may also significantly shape their parents’ psychological adjustment and behaviors. However, most of the existing research in literature on maternal parenting stress has used cross-sectional designs that provide only a snapshot of parenting stress at one moment in time; fewer studies have investigated maternal parenting stress longitudinally. This is surprising if we consider that parenting stress appears to be relatively stable during the preschool period, suggesting that stressed mothers tend to remain stressed, and that parenting stress may accumulate across different child developmental periods, creating the increased risk of poor outcomes for both parenting and child functioning [55]. Therefore, a better understanding of the trends in maternal parenting over the course of early childhood would have important implications not only for understanding the development of children’s social–emotional and emotional–behavioral difficulties and testing directional effects but also for the design of effective early preventive and intervention programs. 

### 1.3. The Present Study

In light of these considerations, the general purpose of the present study was to investigate the associations between maternal parenting, in terms of positive parenting and parenting stress, and preschoolers’ psychosocial adjustment, in terms of social–emotional competence and emotional–behavioral difficulties, both cross-sectionally and longitudinally, at a distance of two years. In particular, we tested an AutoRegressive Cross-Lagged (ARCL) model that incorporated the transactional nature of the associations between mothers’ parenting and children’s skills and behaviors over time (see Figure 1). The model takes into account initial levels of these dimensions and focuses on change in the longitudinal associations. More specifically, we formulated the following hypotheses:

**H1.** 
*Higher levels of children’s social–emotional competence at T1 would be predicted by lower levels of children’s emotional–behavioral difficulties and maternal parenting stress at T0 and by higher levels of positive maternal parenting at T0;*


**H2.** 
*Higher levels of children’s emotional–behavioral difficulties at T1 would be predicted by lower levels of children’s social–emotional competence and maternal positive parenting at T0 and by higher levels of maternal parenting stress at T0;*


**H3.** 
*Higher levels of positive maternal parenting at T1 would be predicted by lower levels of maternal parenting stress and children’s emotional–behavioral difficulties at T0 and higher levels of children’s social–emotional competence at T0;*


**H4.** 
*Higher levels of maternal parenting stress at T1 would be predicted by lower levels of maternal positive parenting and children’s social–emotional competence at T0, and higher levels of children’s emotional–behavioral difficulties at T0.*


**Figure 1 ijerph-19-13750-f001:**
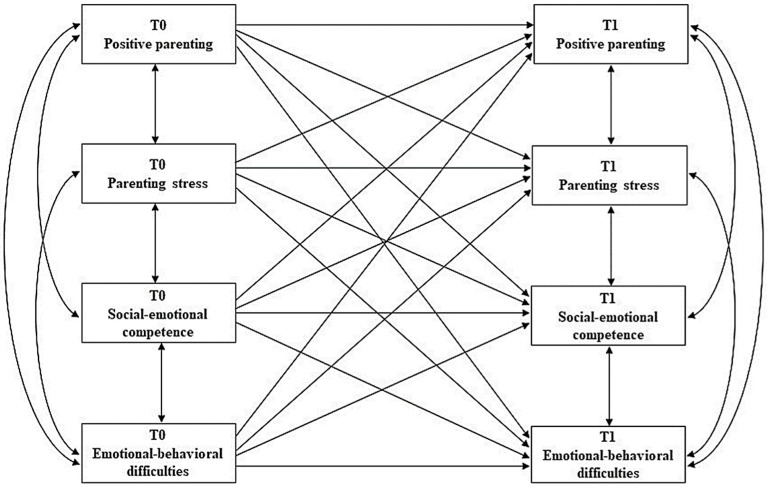
Hypothesized autoregressive cross-lagged model.

## 2. Methods

### 2.1. Participants

The families in this study were participants in a longitudinal study on family socialization and children development. Two psychologists collected the data, owned by the University of Palermo, between February 2019 and April 2021 in a kindergarten in the city of Palermo (Southern Italy) that was part of a project on combating educational poverty. In this study, we examined two waves of data. Participants were 53 Italian mothers living in the South of Italy, aged between 24 and 47 years (M = 35.30, SD = 5.28) at T0. Most of them (93%) were of Italian nationality, and 7% of other nationalities (Romania, China and Dominican Republic); with regard to education level, 9% had a primary school education, 45% had a middle school education, 20% had a high school education and 26% had a university education; with regard to occupational status, 62% were housewives, and 38% were workers; 91% were married or coliving, 9% separated or divorced. The children for whom they responded ranged in age between 3 and 6 years (M = 4.48, SD = 0.84) at T0; 51% of them were females; all children were born in Italy; 28% were an only child, 38% had a brother or sister, and 34% had more than two siblings. Children’s fathers aged between 26 and 49 years (M = 38.08, SD = 5.78). Most of fathers (90%) were of Italian nationality, and 10% of other nationalities (Romania, China, Dominican Republic, Venezuela and Senegal); with regard to education level, 10% had a primary school education, 41% had a middle school education, 41% had a high school education, and 8% had a university education; with regard to occupational status, 26% were unemployed, and 74% were workers. Families in which both parents were of Italian nationality were 90%, families in which both parents were of other nationality than Italian were 8%, and families in which only one parent was of Italian nationality were 2%. 

### 2.2. Procedure

Mothers were recruited in the kindergarten attended by their children in Palermo city (the south of Italy). Those who were interested in participation were informed about the purpose of the research, the voluntary nature of participation and the anonymity of responses. All mothers received and signed informed consent. Both at the baseline (T0) and after two years (T1), they were individually administered the questionnaires, which were delivered in a sealed and anonymous envelope, through their children’s teachers, with the request that they be completed at their home. 

The privacy and anonymity of their answers were guaranteed, and the research obtained the authorization of the local ethics committee. The present study followed the ethical standards of the 1964 Helsinki Declaration and its later amendments or comparable ethical standards. 

### 2.3. Measures

Maternal Positive Parenting. The subscale Positive Parenting, adapted for the current study, of the Alabama Parenting Questionnaire—Preschool Revision (APQ-PR) [56]; Italian adaptation: [57] was used. This subscale consists of 7 items that describe a series of educational and disciplinary behaviors for which the mother has to estimate the frequency of occurrence in ordinary interactions with her child. The Positive Parenting subscale refers, in particular, to mothers’ warm manifestations, reinforcement of their child’s positive behaviors, attention and participation in shared activities (e.g., “I praise my son/daughter when he/she behaves well”). Items were rated on a 5-point Likert scale, ranging from 1 (Never) to 5 (Always). In the present study, this subscale had good internal consistency: Cronbach’s α was 0.69 at T0, and 0.74 at T1.

Maternal Parenting Stress. A scale derived from the Parenting Stress Index—Short Form (PSI-SF) [41]; Italian adaptation: [58] was used. This scale consists of 15 items, articulated in 3 subscales, each consisting of 5 items: (a) Parental Distress, which measures the sense of incompetence about raising of child, the conflict with the partner, the lack of social support and the stress associated with restrictions resulting from the parental role (e.g., “To meet the needs of my child I realize that I sacrifice my life more than I expected”); (b) Parent-Child Dysfunctional Interaction, which reflects the negative feelings related to the expectations towards the child and the confirmation or not of the role as a parent in the relationship with the child (e.g., “I feel that my child does not like me and that he/she does not want to be near me”); (c) Difficult Child, which indicates mother’s perception of child characteristics in terms of temperament, demanding or provocative behaviors, not collaborative and picky (e.g., “My child cries and agitates much more than most children”). Items were rated on a 5-point Likert scale, ranging from 1 (Disagree) to 5 (Strongly agree). In the present study, a total score of parenting stress was obtained by computing the mean of single items scores; the scale had good internal consistency: Cronbach’s α was 0.72 at T0, and 0.87 at T1.

Children’s Social–Emotional Competence. A scale derived by the Social Competence Scale (SCS) [59] was used. This scale consists of 8 items, divided into three subscales: (a) Emotion Regulation, which evaluates the ability to control actively and effectively emotions and inhibit inappropriate behaviors (five items, e.g., “He/she can control his/her mood when he/she disagrees with someone”); (b) Prosocial/Communication Skills, which evaluates the ability to initiate positive interactions with peers and to communicate effectively with others (two items, e.g., “He/she is able to give suggestions and express his/her opinions without being overbearing”); (c) Understanding and Respect of the Rules, which evaluates the ability to recognize, understand and respect the defined rules (one item: “He/she understands and respects the rules”). Items were rated on a 5-point Likert scale, ranging from 1 (Never) to 5 (Always). In the present study, a total score of social–emotional competence was obtained by computing the mean of single items scores; the scale had good internal consistency: Cronbach’s α was 0.77 at T0, and 0.88 at T1.

Children’s Emotional–Behavioral Difficulties. Four subscales of The Strengths and Difficulties Questionnaire (SDQ) [60]; Italian validation: [61] were used: (a) Hyperactivity, which evaluates the presence of difficulties related to attention (3 items, e.g., “He/she is easily distracted, unable to concentrate”); (b) Emotional Symptoms, which evaluates the presence of difficulties in managing situations that can be emotionally stressful (5 items, e.g., “He/she is nervous or uncomfortable in new situations, he/she feels unsure of himself/herself”; (c) Conduct Problems, which evaluates the presence of conduct problems of an externalizing type (5 items, e.g., “He/she argues with other children or intentionally annoys them”); and (d) Peer Problems, which evaluates the presence of problems in the relationship with peers (5 items, e.g., “He/she is lonely, tends to play alone”). Items were rated on a 5-point Likert scale, ranging from 1 (Not true) to 5 (Always true). In the present study, a total score of emotional–behavioral difficulties was obtained by computing the mean of single items scores; the scale had good internal consistency: Cronbach’s α was 0.80 at T0, and 0.86 at T1.

### 2.4. Plan of Data Analysis

Preliminary analyses were conducted to assess study variables univariate distribution. A bivariate correlation analysis was also performed to evaluate the extent to which each variable was related to each other at T0 and T1. 

To analyze the longitudinal associations among the study variables, an ARCL model through structural equation modeling (SEM) was run using Mplus 7 [62]. Such a model is useful in establishing the causal ordering within a single variable over time, as well as the causal relationships between two or more variables over time and the within-time correlations between each construct. All variables included in the model were specified as observed variables. The adequateness of the proposed model was evaluated by inspecting the χ^2^, the root mean square error of approximation (RMSEA), and the comparative fit index (CFI). A nonsignificant χ^2^, RMSEA of 0.05 or lower, and CFI equal to or greater than 0.95 are indicative of an excellent fit [63]. Robust Maximum Likelihood estimation (MLR) method was used to adjust for non-normality distribution. Missing data were handled using full information maximum likelihood (FIML) [64]. We tested the model, which included (a) stability coefficients for all constructs (i.e., autoregressive paths), (b) within-time correlations between the variables and (c) cross-lagged paths between each of the constructs. Equality constraints were imposed on covariances at each time point.

## 3. Results

### 3.1. Preliminary Analyses

Means, standard deviations, skewness and kurtosis of study variables and Pearson correlation coefficients are displayed in Table 1. Variables were normally distributed, falling approximately in the range of −1 to +1 considered acceptable to indicate univariate normality [65], with the exception of children’s emotional–behavioral difficulties (at T0 and T1), maternal positive parenting (at T1) and maternal parenting stress (at T1). The examination of cross-time correlations within each variable showed a good level of temporal stability, averaging 0.35 for maternal positive parenting and 0.52 for maternal parenting stress.

### 3.2. Autoregressive Cross-Lagged Model

The ARCL model fit the data well, χ^2^(6) = 3.37, *p* = 0.76, CFI = 1.00, RMSEA = 0.00. Figure 2 reports the standardized solution of parameter estimates; for clarity purposes, only significant parameter estimates are reported. As for the within-time correlations between different variables, the results showed that maternal parenting stress was negatively and significantly associated with children’s social–emotional competence and positively and significantly related to children’s emotional–behavioral difficulties; mothers’ positive parenting was positively and significantly related to their children social–emotional competence; finally, children’s social–emotional competence and emotional–behavioral difficulties were negatively and significantly associated. In terms of cross-lagged relations, maternal positive parenting at T0 negatively predicted maternal parenting stress at T1; and maternal parenting stress at T0 negatively predicted children’s social–emotional competence at T1.

## 4. Discussion

The current study was aimed at investigating the associations over time between maternal parenting and children’s adaptation. In particular, this longitudinal research examined the associations of both maternal positive parenting and maternal parenting stress with social–emotional competence and emotional–behavioral difficulties in preschool children. The results for each hypothesis are presented and discussed separately below.

Firstly, the prediction that higher levels of children’s social–emotional competence at T1 would be predicted by lower levels of children’s emotional–behavioral difficulties and maternal parenting stress at T0, and by higher levels of positive maternal parenting at T0, was only partially confirmed. Our results provided evidence that maternal parenting stress at T0 negatively predicted children’s social–emotional competence at T1, whereas the effects of maternal positive parenting and children’s emotional–behavioral difficulties at T0 were not significantly associated with children’s social–emotional competence at T1. These findings are in line with the scientific literature highlighting the negative impact of maternal stress on the development of the children, especially in the social–emotional domain [66]. Several authors have argued that high levels of perceived stress may affect parents’ interpretations of children behaviors and may lead parents to difficulties in appropriately managing their own moods and to show excessive reactions [67,68]. These behaviors, in turn, may negatively impact children’s social–emotional competence. Instead, in the model tested in the current study, maternal positive parenting at T0 did not predict children’s social–emotional competence at T1. However, it should be noted that maternal positive parenting and children’s social–emotional competence were positively associated at T1. Therefore, the association between these variables was significant when considered transversally and in a specific moment (T1) even though it was not significant in the long term. Moreover, children’s emotional–behavioral difficulties at T0 did not predict children’s social–emotional competence at T1 in the model even though these two variables are significantly and negatively correlated. The absence of significant predictive effects between these variables in the model it is possibly due to the small sample involved in our study.

Secondly, the hypothesis that higher levels of children’s emotional–behavioral difficulties at T1 would be predicted by lower levels of children’s social–emotional competence and maternal positive parenting at T0 and by higher levels of maternal parenting stress at T0 was not confirmed since these associations were not statistically significant. In other words, maternal parenting stress and positive parenting as well as children’s social–emotional competence at T0 did not predict the presence of emotional–behavioral problems in children at T1. These findings are not in line with those studies highlighting that there is a significant link between negative maternal parenting style, characterized by stress, poor supervision, and hard discipline, and children’s behavioral difficulties [69,70]. However, a further look at the results showed that maternal parenting stress at T0 was positively and significantly correlated with children’s emotional–behavioral difficulties at T1. The absence of significant predictive effects in the model in this case might also be due to the limited numbers of participants in the study. 

Not even the third hypothesis was confirmed by our findings since the associations between emotional–behavioral difficulties, children’s social–emotional competence and parenting stress at T0 with positive parenting at T1 were not significant. In general, it seems that maternal positive parenting for our participants neither predicted nor was associated with other the variables in the study. The only predictor of maternal positive parenting at T1 was maternal positive parenting at T0. Therefore, the current research seems to provide evidence for the stability in the long-time of maternal positive parenting. Our findings have shown that a relationship in which the mothers offer support, protection and recognition to the children is maintained over time and does not suffer the negative effects either of maternal parenting stress or of children’s emotional–behavioral difficulties. Some longitudinal studies have already examined the stability over time of positive parenting behaviors [71,72]. In particular, Landry and colleagues [72] observed a considerable stability in maternal responsivity and positive parenting over the first 4 years of life, while Dallaire and Weinraub [71] have provided evidence that sensitive and stimulating parenting behaviors display considerable and increasing stability over the first years of life.

Finally, the fourth hypothesis was partially confirmed. In particular, the maternal positive parenting at T0 predicted negatively the maternal parenting stress at T1, while the effects of both social–emotional competence and emotional–behavioral difficulties at T0 were not significant on parenting stress at T1, even if emotional–behavioral difficulties at T0 were positively related with parenting stress at T1. These findings highlight that maternal positive parenting can be considered as a protective factor for the onset of parenting stress over time. Maybe mothers that are able to promote supportive interactions with their children, characterized by caring behaviors, warmth, and responsiveness, contribute in this way to create a fertile ground to prevent future manifestations of parenting stress [73,74]. The levels of mothers’ perceived stress may be also negatively affected by some behavioral features of the children like the presence of behavioral and emotional difficulties, which can make the task of the parents even more difficult, although in our model they did not have a predictive power.

When considered together, these findings, even though they do not confirm all our initial hypotheses, underline that parenting stress is a significant risk factor for the impaired development of children’s social–emotional competence over time. Developmental literature has traditionally pointed out the harmful effects of perceptions of stress among caregivers. According to the attachment theory [75], stress could lead caregivers to behave inattentively, to not provide a secure base for children to explore the environment or respond adequately to their needs. As a consequence, children may develop unsecure internal working models that may affect their social–emotional competence. Scientific literature on mother–child attachment has shown that there is a significant correlation between maternal attachment patterns and social–emotional competence in children [76,77]. Recent studies have investigated the link between experiences of insecure attachment characterized by high levels of maternal stress and poor internalization of the strategies of emotional regulation and poverty of social skills by the child [78,79]. Children with insecure attachment relationships tend to develop a model of self as incompetent [80], and their experiences with an insensitive caregiver may influence their beliefs and expectations about future relationships and their social–emotional competence [81]. 

Another interesting element that our study seems to reveal is the beneficial effect of maternal positive parenting on maternal parenting stress over time (and not at the same time point). Therefore, maternal positive parenting acquires an adaptive value in a longitudinal perspective, probably because it creates a familiar or relational atmosphere that over time makes mothers increasingly able to manage potentially stressful situations in a constructive way, reducing their impact and perception of stress.

A further matter highlighted by our research is that there is, at least in our participants, a certain stability in the time of some behaviors like maternal positive parenting, maternal parenting stress, children’s social–emotional competence and emotional–behavioral difficulties. In particular, the stability over time of positive behaviors, like maternal positive parenting or children’s social–emotional competence, can be considered a protective factor for the children’s development. As a consequence, preventive interventions addressed to mothers and children should focus on supporting positive parenting behaviors as well as on improving children’s social and emotional skills in order to increase children’s competences and reduce their problematic behaviors over time [71].

Overall, the findings should be interpreted with caution due to the strengths of the associations between the study variables that are weak/moderate even when statistically significant (i.e., ranging from −0.20 to −0.45). It is likely that the strengths of these associations are affected by the limited size of the sample, which limited the significance of effects.

The results of this study should be also considered in light of other limitations. First, we focused only on maternal perceptions to detect the social–emotional competence and emotional–behavioral difficulties of children. Mothers are often considered the best informers of their children’s behavior [82]. However, it would be necessary to investigate the points of view of fathers and other reference figures (such as teachers, peers, brothers) to get a more complete and accurate picture of the child profile. Fathers’ involvement is necessary especially following the cultural and social changes of recent years, as they are increasingly present in family life [83]. Furthermore, fathers’ point of views could add new elements to the concept of stress parenting and, therefore, provide unexplored research insights into factors that put parenting at risk and possible intervention strategies for the family system. Secondly, although this study has a longitudinal structure that monitors maternal parenting and child’s social–emotional competence and behaviors, it is necessary to explore more deeply the predictors of these factors in the long term considering them in their interaction and synergy. Finally, as we said above, the sample of the research was small; moreover, it was mostly composed of Italian mothers coming from an urban community. Therefore, it is unclear to what extent these findings generalize to other populations. Future studies should include larger and more diversified samples that would be appropriate for more sophisticated statistical analyses and that would allow for more accurate comparisons across different ethnic groups.

## 5. Conclusions

Despite these deficiencies, the current paper may contribute to advancing research and practice related to parenting practices and children’s adjustment. With regard to the field of research, the added value of our study consists in highlighting the importance of longitudinal research and these topics, since longitudinal data provide more and different information from cross-sectional findings with regard to long-term associations between variables. Longitudinal findings may suggest the existence of developmental pathways that can be tested in further research. In particular, our findings seem to support predictions about the protective role of maternal positive parenting in reducing the occurrence of maternal parenting stress, as well as about the importance of reducing maternal parenting stress in order to promote higher levels of children’s social–emotional competence. These predictions can be tested in future studies. 

With regard to the field of practice, the added value of our study consists in emphasizing the essential function that could be promoted by means of psychological interventions centred on mother-child interactions and positive parenting like parent training programs. Such intervention programs could also focus on developing a secure attachment between mother and child, as this reduces the negative effects of parenting stress on children’s social–emotional competence [84]. Specifically, practitioners’ work could stimulate mothers to build more appropriate interactions with their children, also helping them to learn new strategies to manage their parenting issues. One the one hand, this could decrease maternal parenting stress and increase mothers’ sense of effectiveness as parents as well as their confidence in their own abilities; on the other hand, psychological intervention could promote children’s social–emotional competence and mitigate the risk of occurrence of emotional–behavioral difficulties.

## Figures and Tables

**Figure 2 ijerph-19-13750-f002:**
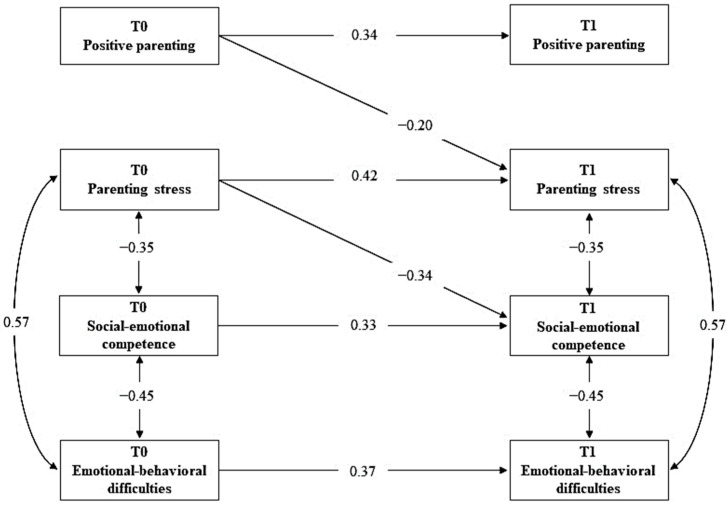
Statistical autoregressive cross-lagged model. Standardized solution.

**Table 1 ijerph-19-13750-t001:** Means (M), standard deviations (SD), skewness (S), kurtosis (K), and Pearson correlation coefficients of study variables at T0 and T1.

	M	SD	S	K	1	2	3	4	5	6	7	8
1. T0 Social–emotional competence	3.66	0.64	−0.64	1.67	-							
2. T1 Social–emotional competence	3.70	0.71	−0.53	−0.41	0.42 **	-						
3. T0 Emotional–behavioral difficulties	1.91	0.48	1.48	4.34	−0.43 **	−0.28 *	-					
4. T1 Emotional–behavioral difficulties	1.88	0.57	2.36	8.07	−0.15	−0.56 **	0.47 **	-				
5. T0 Positive parenting	4.90	0.31	−0.74	1.37	0.12	0.17	−0.11	0.02	-			
6. T1 Positive parenting	5.52	0.44	−1.45	2.99	0.15	0.43 **	−0.20	−0.27	0.35 *	-		
7. T0 Parenting stress	1.59	0.47	1.03	1.63	−0.33 *	−0.41 **	0.56 **	0.43 **	−0.11	−0.26	-	
8. T1 Parenting stress	1.55	0.60	2.04	4.93	−0.22	−0.51 **	0.35 *	0.63 **	−0.23	−0.25	0.52 **	-

* *p* < 0.05; ** *p* < 0.01.

## Data Availability

All data sets are available upon request from the corresponding author.

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
