# Peer review of "Maternal Parenting and Preschoolers’ Psychosocial Adjustment: A Longitudinal Study"

_ijerph, 2022, doi:10.3390/ijerph192113750_

Round 1

Reviewer 1 Report

1.  At line# 335-367, did the analytical model included demographic variables as control variables in data analysis? If not, inclusion of demographic variables in a full model may lead to different results.

2.  In Figure 2, what are the interpretations of the reported numerical values regarding the strengths of associations between variables? The discussion section will need to include such interpretation and its influence on the findings.

Author Response

Response to Reviewer 1 Comments

Point 1: At line #335-367, did the analytical model included demographic variables as control variables in data analysis? If not, inclusion of demographic variables in a full model may lead to different results.

Response 1: Many thanks for the suggestions. The analytical model did not include demographic variables as control variables; we decided not to include them for the following reasons. Child’s age and gender were preliminarily included in the model as covariates influencing all study variables at T0 (Mplus output file of this model is attached to the revision), but neither of them was a significant predictor. Therefore, in order to reduce the complexity of the model (especially, in consideration of the small sample size), we decided to test a model without covariates.

Point 2: In Figure 2, what are the interpretations of the reported numerical values regarding the strengths of associations between variables? The discussion section will need to include such interpretation and its influence on the findings.

Response 2: We thank the Reviewer for having proposed this integration to the text. We added the interpretation of the strengths of associations between variables in the Discussion. 

Reviewer 2 Report

Dear authors,

Please consider and deal with the aspects mentioned in the following:

Introduction section:

Why the article Maternal Parenting Stress and Preschoolers’ Social-Emotional Competence and Behavioural Difficulties: A Variable- and Person-Centred Approach was not cited by the authors?

https://cab.unime.it/journals/index.php/JCDP/article/view/3375

Methods section:

Plese give more details about the ‘’longitudinal research on family socialization and children development’’ (who collected the data, who is the owner of the data, the period of the longitudinal study - nomination years, the motivation for choosing the type of sample, the longitudinal evolution of the sample)

Disscussion section:

Please develop a specific paragraph that emphasizes the added value of the article. The limitation is indeed primarily due to the small number of participants in the study, but the results can contribute to the development of tools to be tested in further research, and this must be mentioned

After the authors' contribution, please add:

Funding

Institutional Review Board Statement

Informed Consent Statement:

Informed consent

Data Availability Statement

Author Response

Response to Reviewer 2 Comments

Point 1: 

Introduction section

Why the article Maternal Parenting Stress and Preschoolers’ Social-Emotional Competence and Behavioural Difficulties: A Variable- and Person-Centred Approach was not cited by the authors?

https://cab.unime.it/journals/index.php/JCDP/article/view/3375  

Response 1: Thanks for this remark. In the revised manuscript, we cited the above mentioned article in the Introduction section. The new source cited in the text has also been included in the References section.

Point 2: 

Methods section

Plese give more details about the ‘’longitudinal research on family socialization and children development’’ (who collected the data, who is the owner of the data, the period of the longitudinal study - nomination years, the motivation for choosing the type of sample, the longitudinal evolution of the sample).

Response 2: Thank you for this suggestion. In the Method section we have added this information.

Point 3: 

Discussion section

Please develop a specific paragraph that emphasizes the added value of the article. The limitation is indeed primarily due to the small number of participants in the study, but the results can contribute to the development of tools to be tested in further research, and this must be mentioned.

Response 3: Many thanks for this comment. Now, we have highlighted the added value of our article on light of your suggestion.

Point 4: After the authors' contribution, please add: Funding, Institutional Review Board Statement, Informed Consent Statement: Informed consent, Data Availability Statement.

Response 4: We have added the requested information after the authors’ contribution.